# EEGPT: Unleashing the Potential of EEG Generalist Foundation Model by Autoregressive Pre-training

## Abstract

Electroencephalogram (EEG) signals are pivotal in providing insights into spontaneous brain activity, highlighting their significant importance in neuroscience research. However, the exploration of versatile EEG models is constrained by diverse data formats, outdated pre-training paradigms, and limited transfer learning methods, only leading to specialist models on single dataset. In this paper, we introduce EEGPT, the first generalist EEG foundation model designed to address these challenges. First, we propose an electrode-wise modeling strategy that treats each electrode as a fundamental unit, enabling the integration of diverse EEG datasets collected from up to 138 electrodes, amassing 37.5M pre-training samples. Second, we develop the first autoregressive EEG pre-trained model, moving away from traditional masked autoencoder approaches to a *next signal prediction* task that better captures the sequential and temporal dependencies of EEG data. We also explore scaling laws with model up to 1.1B parameters — the largest in EEG research to date. Third, we introduce a multi-task transfer learning paradigm using a learnable electrode graph network that is shared across tasks, which for the first time confirms multi-task compatibility and synergy. As the first generalist EEG foundation model, EEGPT shows broad compatibility with various signal acquisition devices, subjects, and tasks. It supports up to 138 electrodes and any combination thereof as input. Furthermore, we simultaneously evaluate it on 5 distinct downstream tasks across 12 benchmarks. EEGPT consistently outperforms existing specialist models across all downstream tasks, with its effectiveness further validated through extensive ablation studies. This work sets a new direction for generalist EEG modeling, offering improved scalability, transferability, and adaptability for a wide range of EEG applications. Both the training code and model checkpoints will be publicly available.

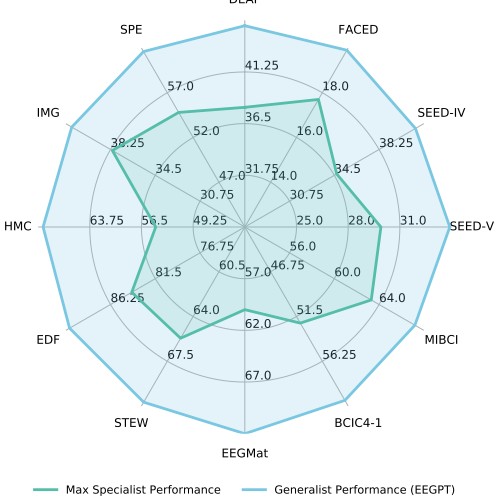

**Figure 1:** EEGPT, as a generalist model, significantly outperforms dataset-specific specialist models across 12 benchmarks spanning 5 tasks. It strongly demonstrate the versatility and transferability.

# 1 INTRODUCTION

Electroencephalogram (EEG), which captures spontaneous brain activity via electrograms (Biasiucci et al., 2019), could be conceptualized as the language of the brain. Through the analyses of EEG, valuable insights are derived for various applications, including but not limited to emotion recognition (Zhang et al., 2024), motor imagery classification (An et al., 2023), mental workload detection (Wang et al., 2024b) and sleep stage classification (Liang et al., 2023). This breadth of applications underscores the versatility and utility of EEG in neuroscientific research.

Research on EEG downstream tasks has been thriving, yet most studies share a notable characteristic: ***specialization***. For instance, at the data level, a variety of proprietary data formats (Jia et al., 2020; Bashivan et al., 2015) and handcrafted feature extraction (Duan et al., 2013; Yan et al., 2023) techniques have been introduced to enhance the discriminability of domain-specific data. At the model level, various modules and structures are designed and trained for a specific task (Shao et al., 2023), dataset (Wang et al., 2024a), or even individual subjects (Gao et al., 2024). However, a ***generalist*** EEG foundation model is highly anticipated, as it offers broader applicability across various EEG tasks. Moreover, this model improves transfer learning by allowing knowledge from one task to enhance performance on others. Its design also demonstrates greater robustness to data and task variations, leading to better generalization in unseen scenarios.

Although extensive research in the fields of computer vision (CV) (Radford et al., 2021; Dosovitskiy, 2020; Bai et al., 2024) and natural language processing (NLP) (Radford et al., 2019; Achiam et al., 2023; Brown, 2020) has identified three key components for constructing generalist models—data, self-supervised pre-training, and transfer learning paradigms—EEG introduces its own unique and daunting challenges in each of these domains:

**Data Format.** EEG data exhibit significant heterogeneity (Wang et al., 2024a; Saeed et al., 2021), characterized by a variety of systems (*e.g.*, the 10-20 system) and equipment (*e.g.*, Neuroscan) used in data collection. Furthermore, different datasets may employ a diverse number and combination of electrodes based on practical considerations. The inconsistency in data formats across different sources prevents their combined use in the same model for training, making it challenging to develop a generalist for EEG. Therefore, an efficient and scalable strategy for unifying these diverse EEG data format is extremely demanding.

**Self-supervised Pre-training.** Current studies (Yang et al., 2024b; Jiang et al., 2024; Yi et al., 2024) have uniformly employed techniques that mask parts of EEG signals and utilize a bidirectional attention mechanism (Vaswani, 2017) to reconstruct the masked data (*i.e.*, mask autoencoder, MAE). However, they have inevitable limitations in capturing the sequential and temporal dependencies inherent in time-based data such as language and EEG. Given the gradual obsolescence of MAE architectures in NLP (Zhao et al., 2023; Minaee et al., 2024), it is essential for the EEG field, which shares similar temporal dynamics with language, to reconsider its current pre-training paradigms.

**Transfer Learning.** Current pre-trained EEG models are generally fine-tuned for specific datasets, resulting in specialists in narrow domains. However, in CV and NLP, many pre-trained models (Touvron et al., 2023; Yang et al., 2024a; Wang et al., 2023) have achieved remarkable generalizability through more adaptable and efficient knowledge transfer learning methods. These models support multiple tasks and promote beneficial synergies. Compared to models specialized in single tasks, they exhibit enhanced and broader capabilities, facilitating a more thorough utilization of pre-trained knowledge. However, advanced transfer learning method remains underexplored in the EEG field.

In this paper, we propose EEGPT, a generalist EEG foundation model offering extensive versatility. Specifically, it seamlessly adapts and encodes signals collected by nearly all popular EEG acquisition devices. It accommodates signals from up to 138 electrodes, supporting various configurations and combinations. Moreover, EEGPT is capable of simultaneously processing and analyzing data from nearly all prevalent downstream tasks within a single model, and it is highly scalable to new tasks. The training recipe for EEGPT significantly diverges from previous paradigms, with its novelty encapsulated in five distinct "***firsts***":

1) For data format, we propose the **first electrode-wise modeling strategy**. It deconstructs the signals electrode by electrode. Each electrode serves as a fundamental unit for subsequent training. Although the sets of electrodes differ across various datasets, this strategy consistently translates into an electrode-conditioned temporal modeling task. Leveraging this compatibility and unification, we extensively collect a total of 37.5M pre-training samples.

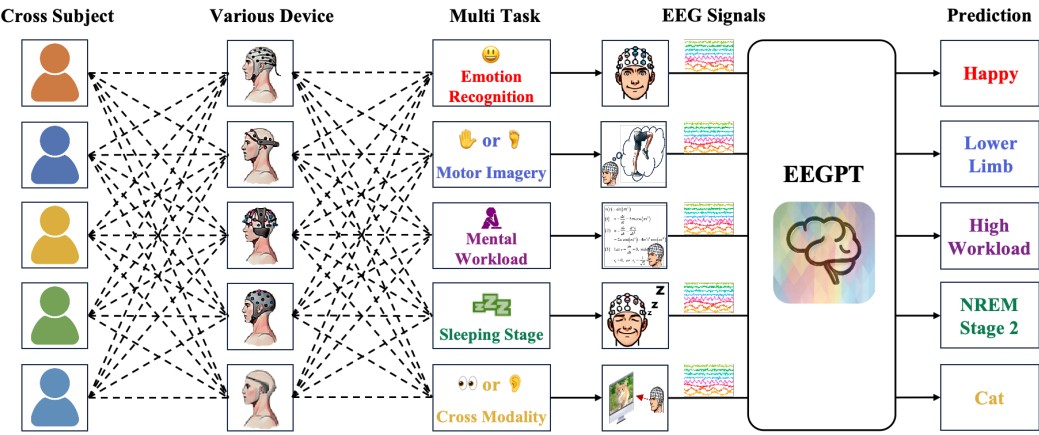

**Figure 2:** The versatility of EEGPT is reflected in the broad compatibility with subjects, signal acquisition devices, and tasks. EEG signals from various subject, using various device, and performing various task can be characterized and analyzed effectively within one model, and it exhibits remarkable scalability.

2) For self-supervised pre-training, we propose the **first autoregressive EEG pre-trained model**, seamlessly accommodating the sequential and temporal dependencies inherent in EEG data. Compared to MAE, the pre-trained model engages in a more intuitive yet challenging task of "*next signal prediction*". Based on 37.5M training samples, we split approximately 1B tokens to conduct pre-training across four scales (*i.e.*, Base, Huge, Large, and Giant). To the best of our knowledge, It is the **first exploration and validation of the scaling laws for autoregressive architectures in the EEG domain**. Besides, EEGPT-Giant has achieved about 1.1B parameters, marking it as the **first model in the EEG field to exceed the billion-parameter threshold**.

3) For transfer learning, we propose a learnable graph network, with electrodes as nodes, is concurrently shared across multiple tasks. Task-specific node activation patterns are adaptively determined by corresponding input data format. Leveraging the robust temporal representations learned from electrode-conditioned pre-training, the electrode graph serves as a spatial supplement by integrating information from multiple electrodes. The whole framework is designed with a progressive spatiotemporal decoupling. We collect data from 12 benchmarks for multi-task learning instead of traditional single-task fine-tuning. Interestingly, the tasks demonstrate a notable mutual enhancement. It establishes EEGPT as the **first generalist EEG model for multi-task compatibility and synergism**.

Based on these designs, our contributions are summarized as follows:

- **Electrode-wise Modeling Strategy**. We introduce a novel electrode-wise strategy for EEG data integration, treating each electrode as a fundamental unit across various datasets. This approach enables uniform handling and scalability in data processing. Benefiting from this method, EEGPT can support up to 138 electrodes and their arbitrary combinations, offering flexibility and applicability far beyond existing models.

- **Autoregressive EEG Pre-trained Model**: We introduce the first autoregressive pre-trained EEG model. Compared to traditional MAE techniques, it more naturally and efficiently captures the sequential and temporal dynamics inherent in EEG data. The scaling laws for data and model size in the autoregressive framework have been effectively validated.

- **Multi-task Transfer Learning Paradigm**: Building upon a learnable task-shared graph network, EEGPT is the first generalist model to exhibit confirmed multi-task compatibility and synergism. Significant mutual enhancement across tasks are demonstrated through multi-task transfer learning.

- **Comprehensive Quantitative and Qualitative Experiments**. EEGPT demonstrated superior performance across 12 datasets encompassing 5 tasks, surpassing both pretrain-then-finetune and training-from-scratch predominant specialist baselines. The effectiveness of our proposed method is further validated by extensive qualitative analyses.

## 2 METHOD

In this section, we elaborate on the comprehensive framework of EEGPT. A detailed framework is illustrated in Figure 3. We begin by representing a multi-electrode EEG signal as $x \in \mathbb{R}^{E_i \times T \times C}$, where $E_i$ represents the number of electrodes. For the entire signal, we segment it into $T$ one-second intervals. Each interval is represented by $C$ uniformly sampled points from the original signal.

### 2.1 AUTOREGRESSIVE TIME SERIES MODELING

In this stage, we aim to develop a comprehensive and detailed self-supervised learning paradigm. It is designed to accurately and efficiently capture the intrinsic temporal variations in EEG signals. Furthermore, we plan for it to be electrode-conditioned, which will facilitate the discernment of both disparities and similarities across different electrodes, enhancing its effectiveness in diverse scenarios.

**Electrode-wise Modeling Strategy.** We compile the pre-training dataset $X = \{x_1, x_2, \ldots, x_N\}$ by aggregating data from multiple sources. Each sample $x_i \in \mathbb{R}^{E_i \times T \times C}$ corresponds to a set $\mathcal{E}_i$ containing $E_i$ electrodes. To discern the distinctive patterns specific to each electrode in EEG signals, we introduce a structured reorganization function $\mathcal{R}(\cdot)$. Specifically, we segment each $x_i$ based on individual electrodes, denoted as $\mathbf{x}_i^e \in \mathbb{R}^{T \times C}$. Consequently, EEG recordings from disparate sources that share identical electrodes are grouped together:

$$\mathcal{R}(X) = \left\{ D_e \mid e \in \bigcup_{i=1}^{N} \mathcal{E}_i \right\} \tag{1}$$

where $D_e$ is the grouped collection of all data segments $\mathbf{x}_i^e$ from electrode $e$ across all samples that include electrode $e$:

$$D_e = \{\mathbf{x}_i^e \mid e \in \mathcal{E}_i, \, i = 1, 2, \ldots, N\} \tag{2}$$

The size of $\mathcal{R}(X)$ corresponds to the count of unique electrodes present in $X$. To distinguish between different electrodes, we introduce a trainable electrode vocabulary $v_E \in \mathbb{R}^{|R(X)| \times C}$. All elements in $D_e$ share the same electrode embedding $v_E^e$. This embedding is served as condition and then concatenated along the sequence dimension to $x_i^e$. For simplicity, we consistently refer to the concatenated sequence as $x_i^e$:

$$x_i^e = [v_E^e || x_i^e] \in \mathbb{R}^{(T+1) \times C} \tag{3}$$

where $\|$ signifies the concatenation operation. Hence, signals from various domains and electrodes have been standardized into a highly scalable format. The chronological sequences $x_i^e$, which contains $T+1$ EEG "*tokens*", will serve as the fundamental unit for performing autoregressive reconstruction.

**Autoregressive Reconstruction.** As depicted in Figure 3 (left), each $x_i^e$ is inputted into a shared Electrode Temporal Encoder (ETE), which comprises a series of $L$ identical layers. Each layer contains two sub-layers: the first utilizes a multi-head causal attention mechanism, and the second employs a positionwise fully connected feed-forward network. Specifically, the input sequence $x_i^e$ first undergoes the causal attention process:

$$\text{Attention}(Q, K, V) = \text{softmax}\left(\frac{QK^T}{\sqrt{d}} + M\right) V \tag{4}$$

where $Q$, $K$, and $V$ are queries, keys, and values respectively, all derived from $x_i^e$, and $d$ is the hidden size. $M$ is a causal mask designed to ensure that each token only attend to tokens that are sequentially prior to itself. The output of this sub-layer is then normalized and passed through a residual connection (He et al., 2016). Subsequently, it is fed into feed-forward network, which consists of two linear transformations with a SwiGLU (Dauphin et al., 2017) activation. Finally, the output from ETE is transformed into the corresponding next-token prediction through a simple MLP.

**Training Objective.** Assuming the input is $x$, the reconstructed result is denoted as $\hat{x}$. Without loss of generality, the training objective for the autoregressive model can be formulated as follows:

$$\mathcal{L}(\theta) = \frac{1}{T} \sum_{t=1}^{T} \rho(x_t - \hat{x}_t(x_{<t}; \theta)) \tag{5}$$

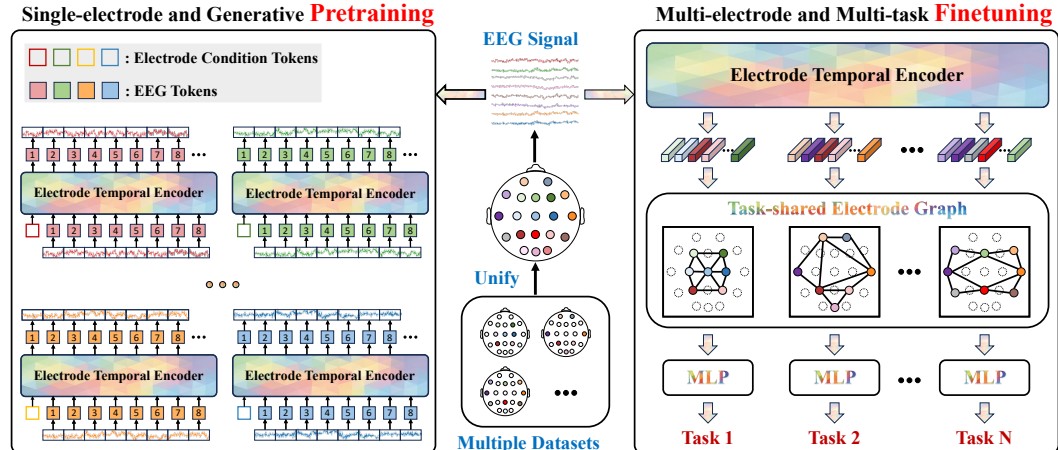

**Figure 3:** Overview of the EEGPT architecture. (left) Autoregressive reconstruction serves as the pre-training objective, with a learnable condition token added at the start to distinguish between electrodes. Each signal token predicts the next token through the Electrode Temporal Encoder (ETE) one-by-one. (right) Electrodes in each dataset are processed through the pre-trained ETE, extracting the final token as the electrode representations, which are then fed into a Task-shared Electrode Graph (TEG) network to integrate spatial information across multiple electrodes. ETE and TEG collectively constitute a progressive spatiotemporal decoupling.

where $\theta$ represents the model parameters that require optimization. The function $\rho(\cdot)$ serves as a distance metric to quantify the discrepancy between the actual and reconstructed values. For LLMs, due to the discrete nature of the vocabulary, Cross-Entropy (CE) is commonly used. Given the inherent continuity of EEG signals, we default to using Mean Squared Error (MSE) in this context.

## 2.2 TASK-SHARED ELECTRODE GRAPH

Through autoregressive pre-training, ETE has effectively captured the temporal characteristics conditioned by electrodes. In this stage, unlike previous pre-trained models which are fine-tuned for individual tasks, we aim to explore a more versatile multi-task paradigm. Specifically, we propose a Task-shared Electrode Graph (TEG) network. This network adaptively activates interactions among various electrodes to simultaneously support multiple tasks.

**Electrode Representation Extraction.** Consider a multi-task dataset defined as $Y = \{y_1, y_2, \ldots, y_M\}$. Each sample $y_j$ belongs to $\mathbb{R}^{E_j \times T \times C}$, where $E_j$ denote the number of electrodes. As illustrated in Figure 3 (right), for each sample $y_j$, a learnable special token $c \in \mathbb{R}^C$ is broadcast across all $E_j$ electrodes and appended to the end of the temporal sequence:

$$y'_j = \left[ y_j \| c \cdot \mathbf{1}_{E_j \times 1} \right] \in \mathbb{R}^{E_j \times (T+1) \times C} \tag{6}$$

Leveraging the unidirectional attention mechanism inherent to autoregressive models, these special tokens facilitate the integration of local information from individual electrodes to synthesize global representations. Specifically, $y'_i$ are then processed by the pre-trained ETE. Notably, the parameters of ETE are frozen during this stage, functioning solely as a feature extraction backbone. Subsequently, electrode representations are derived from the positions of special tokens in the output of ETE:

$$z_j = \text{ETE}(y'_j)[:, -1, :] \in \mathbb{R}^{E_j \times C} \tag{7}$$

Similarly, each sample in $Y$ generates a corresponding $z_j$ that captures comprehensive temporal information. These representations are then input into the proposed TEG network, which is specifically designed to model dependencies among electrodes, integrating spatial information effectively.

**Network Structure.** We initially construct a graph network in which each node represents an electrode utilized during the pre-training stage. The total number of nodes is denoted by $|R(X)|$. Benefiting from the comprehensive data coverage in the pre-training, these nodes include nearly all electrodes commonly employed, encompassing those found in $Y$. Each node is represented by a

learnable vector of length $C$. They form a fully interconnected graph $\mathcal{G} \in \mathbb{R}^{|R(X)| \times C}$. For each $z_j$, the electrodes it comprises are mapped to a subgraph $\mathcal{G}_j$ within $\mathcal{G}$. Upon the introduction of $z_j$ into the network, only the nodes contained within the subgraph $\mathcal{G}_j$ are activated. This activation process involves updating the representations of these specific nodes by adding the corresponding elements from $z_j$:

$$\mathcal{G} = \mathcal{G} + \mathbf{I}_{\mathcal{G}_j} \cdot \mathrm{diag}(z_j) \cdot \mathbf{1}^T \tag{8}$$

The function $\mathrm{diag}(z_j)$ converts $z_j$ into a diagonal matrix, facilitating targeted activations that only influence corresponding nodes within $\mathcal{G}_j$. The indicator matrix $\mathbf{I}_{\mathcal{G}_j}$ ensures that updates are confined to these nodes, leaving others unaffected. The updated graph $\mathcal{G}$ facilitates the flow and interaction of spatial information between electrodes through a graph attention mechanism (Veličković et al., 2017):

$$\alpha_{mn} = \frac{\beta_{mn} \cdot \exp\left(\mathrm{LeakyReLU}\left(\mathbf{a}^T[\mathbf{W}h_m \| \mathbf{W}h_n]\right)\right)}{\sum_{k \in \mathcal{N}(m)} \beta_{mk} \cdot \exp\left(\mathrm{LeakyReLU}\left(\mathbf{a}^T[\mathbf{W}h_m \| \mathbf{W}h_k]\right)\right)} \tag{9}$$

where $\alpha_{mn}$ represents the attention coefficient between nodes $m$ and $n$. $\mathbf{W}$ and $\mathbf{a}$ are the learnable mapping weights. $h_m$ is the representation of node $m$ in $\mathcal{G}$. We use $\mathcal{N}(m)$ to denote the neighbor set of node $m$. For the activated subgraph, a masking coefficient $\beta$ is introduced, where $\beta_{mn}$ equals 1 if both $m$ and $n$ are within it, and 0 otherwise. Based on the obtained attention coefficients, the interactions between nodes are as follows:

$$h'_m = \sigma\left(\sum_{n \in \mathcal{N}(m)} \alpha_{mn}\mathbf{W}h_n\right) \tag{10}$$

where $\sigma$ represents the activation function (ReLU (Glorot et al., 2011) in this context). Similarly, the above operation is stacked across $K$ layers, with each layer employing a residual connection and pre-normalization. For various subgraphs $\mathcal{G}_j$ (*i.e.*, different datasets or tasks) within the same batch, unified training is efficiently achieved by constructing corresponding mask matrices $\beta$. This approach allows the model to operate as a multi-task generalist. In terms of output, the graph network pools the representations of nodes within $\mathcal{G}_j$ and subsequently directs them to the relevant task-specific head for either classification or regression.

## 3 EXPERIMENTS

### 3.1 EXPERIMENTAL SETTINGS

**Model Variants.** We have developed four architecture configurations of EEGPT: EEGPT-Base, EEGPT-Large, EEGPT-Huge, and EEGPT-Giant. The parameter counts for these models are as follows: EEGPT-Base is 1.46M, EEGPT-Large is 11.29M, EEGPT-Huge is 183.8M, and EEGPT-Giant is 1.09B. In the case of the ETE and TEG network, they share the same hidden size and number of attention heads. These increments, which approximately scale by an order of magnitude at each level, are achieved by expanding the depth and width of the network. For a more detailed analysis of scaling law, please refer to Sec 3.3.

**Table 1:** Configuration of EEGPT models.

| Configuration | Base | Large | Huge | Giant |
|---|---|---|---|---|
| ETE Layers | 3 | 9 | 12 | 20 |
| TEG Layers | 3 | 3 | 4 | 4 |
| Head Size | 32 | 32 | 64 | 64 |
| Hidden Size | 128 | 256 | 896 | 1,792 |
| Attention Heads | 4 | 8 | 14 | 28 |
| Intermediate Size | 512 | 1,024 | 3,584 | 7,168 |
| Total Parameters | 1.46M | 11.29M | 183.8M | 1.09B |

**Training Details.** We adopt AdamW (Loshchilov & Hutter, 2017) as the optimizer and conduct all training on 8 NVIDIA A800-SXM4-80G GPUs. To enhance training efficiency, we utilize DeepSpeed Zero Optimization Stage 2. During pre-training, all model scales are trained for 3 epochs using a consistent dataset of 37.5M samples, which collectively includes approximately 1B tokens. The batch size and learning rate are set to 4096 and 1e-4, respectively. The maximum duration for pre-training (for EEGPT-Giant) is capped at 20 hours. For multi-task fine-tuning, all model scales are trained for 10 epochs using a consistent dataset of 181K samples. The batch size and learning rate are maintained at 512 and 1e-4, respectively. The maximum training duration for multi-task learning (for

EEGPT-Giant) is limited to 3 hours. In this stage, the pre-trained parameters of ETE are frozen, and only the newly introduced TEG network is actively trained. Please refer to the Appendix for further details.

**Baseline Models.** The baseline models selected for comparison are divided into two distinct categories. The first category encompasses traditional and widely utilized architectures in the EEG domain, such as EEGNet (Lawhern et al., 2018), TSception (Ding et al., 2022), Conformer (Song et al., 2023), and LGGNet (Ding et al., 2023). These models are trained from scratch on the respective datasets without any pre-training. The second category includes cutting-edge pre-trained models, *i.e.*, LaBraM (Jiang et al., 2024) and BIOT (Yang et al., 2024b), They are fine-tuned on the respective datasets using inherited pretrained parameters. More details regarding of the baseline models could be found in the Appendix.

Considering that no existing models have been evaluated on such a diverse range of downstream tasks, we have meticulously reproduced these models using their official code, hyperparameter configurations, and pretrained checkpoints. This reproduction aims to supplement the performance metrics for each task, facilitating a comprehensive comparison. It is important to note that all baseline results are derived from models fine-tuned for specific tasks, indicating that these are individual specialist models. In contrast, the results from EEGPT originate from a single generalist model.

**Evaluation Details.** We evaluate our EEGPT using 12 datasets across 5 distinct tasks, as detailed in Table 2. For Emotion Recognition (ER), we utilize DEAP (Koelstra et al., 2011), FACED (Chen et al., 2023), SEED-IV (Zheng et al., 2018), and SEED-V (Liu et al., 2021). For Motor Imagery (MI) classification, we employ MIBCI (Cho et al., 2017) and BCI Competition IV-1 (Blankertz et al., 2007). For Mental Workload (MW) detection, we select EEGMat (Zyma et al., 2019) and STEW (Lim et al., 2018). For Sleeping Stage (SS) classification, we analyze

**Table 2:** Statistical analysis of 12 evaluation datasets. Our in-house data is denoted by †.

| Task | Dataset | Rate | # Subject | # Electrode | # Sample | # Class |
|------|---------|------|-----------|-------------|----------|---------|
| ER | DEAP | 128Hz | 32 | 32 | 19.2k | 4 |
| | FACED | 1000Hz | 123 | 30 | 27.6k | 9 |
| | SEED-IV | 200Hz | 15 | 62 | 37.6k | 4 |
| | SEED-V | 200Hz | 16 | 62 | 29.2k | 5 |
| MI | MIBCI | 512Hz | 52 | 64 | 10.5k | 2 |
| | BCIC4-1 | 100Hz | 7 | 38 | 1.4k | 2 |
| MW | EEGMat | 500Hz | 34 | 19 | 1.0k | 2 |
| | STEW | 128Hz | 45 | 14 | 3.3k | 3 |
| SS | EDF | 100Hz | 78 | 2 | 19.5k | 5 |
| | HMC | 256Hz | 151 | 4 | 22.6k | 5 |
| CM | IMG† | 1000Hz | 29 | 122 | 7.6k | 5 |
| | SPE | 256Hz | 7 | 64 | 1.3k | 2 |

data from EDF (Kemp et al., 2000) and HMC (Alvarez-Estevez & Rijsman, 2021). For Cross Modality (CM) tasks, we employed IMG, our proprietary dataset, and SPE (Nguyen et al., 2017). All datasets use accuracy as the performance metric. Further descriptions and processing details are available in the Appendix.

Across all datasets, we adopt a cross-subject paradigm. Specifically, we partition each dataset in the multi-task set into training, validation, and test splits using an 8:1:1 ratio, ensuring no overlap of subjects among these splits. To minimize variability, we calculate the average accuracy and standard deviation from results obtained using five distinct random seeds.

## 3.2 PERFORMANCE EVALUATION

Table 3 presents a performance comparison across 12 datasets, illustrating that EEGPT, despite being a generalist model, consistently surpasses specialist models that have been fine-tuned for specific tasks. Specifically, EEGPT-Giant achieves an average accuracy improvement of 5.07% on the ER task, 6.05% on the MI task, 8.50% on the MW task, 11.20% on the SS task, and 5.10% on the CM task compared to the best performances by these specialist models. Moreover, as the model scales, there is a clear and consistent upward trend in performance improvement.

Interestingly, our findings reveal that specialist models with pre-training appear to perform slightly worse than those trained from scratch. One intuitive hypothesis is that current mainstream EEG pre-training models are often based on large-scale seizure data, which exhibits domain discrepancy from typical EEG data used in general downstream tasks. This mismatch likely hampers the efficacy of transfer learning. Nonetheless, EEGPT models demonstrate considerable versatility and effectiveness across a diverse array of tasks, thereby robustly validating its utility and performance.

**Table 3: Evaluation on EEG Benchmarks.** The column "One Model?" indicates whether the results for these benchmarks originate from the same model. The results in **bold** and underline are the best and second-best results, respectively.

| Method | One Model? | Emotion Recognition | | | | Motor Imagery | |
|---|---|---|---|---|---|---|---|
| | | DEAP | FACED | SEED-IV | SEED-V | MIBCI | BCIC4-1 |
| *Specialist Models w/o pre-train* | | | | | | | |
| EEGNet | ✗ | $35.2 \pm 9.4$ | $15.3 \pm 1.3$ | $28.7 \pm 1.5$ | $28.5 \pm 3.2$ | $63.3 \pm 7.2$ | $51.9 \pm 1.5$ |
| TSception | ✗ | $34.3 \pm 8.1$ | $14.0 \pm 1.8$ | $32.2 \pm 3.6$ | $29.9 \pm 7.0$ | $61.4 \pm 6.5$ | $52.2 \pm 1.6$ |
| Conformer | ✗ | $38.0 \pm 8.7$ | $14.1 \pm 3.6$ | $29.6 \pm 2.3$ | $26.5 \pm 1.0$ | $52.6 \pm 3.0$ | $51.6 \pm 1.8$ |
| LGGNet | ✗ | $33.5 \pm 8.5$ | $17.0 \pm 2.7$ | $34.7 \pm 3.5$ | $29.7 \pm 6.3$ | $56.7 \pm 3.7$ | $50.0 \pm 0.4$ |
| *Specialist Models w/ pre-train* | | | | | | | |
| BIOT | ✗ | $35.2 \pm 8.9$ | $17.7 \pm 2.6$ | $32.7 \pm 4.8$ | $28.8 \pm 4.0$ | $53.2 \pm 2.0$ | – |
| LaBraM | ✗ | $34.3 \pm 9.9$ | $15.5 \pm 1.6$ | $29.5 \pm 2.1$ | $26.4 \pm 0.7$ | $50.5 \pm 1.1$ | $50.3 \pm 0.4$ |
| *Generalist Models* | | | | | | | |
| EEGPT-Base | ✓ | $41.4 \pm 2.7$ | $16.9 \pm 1.3$ | $34.0 \pm 1.7$ | $28.1 \pm 0.9$ | $62.2 \pm 2.8$ | $56.9 \pm 1.6$ |
| EEGPT-Large | ✓ | $42.5 \pm 3.8$ | $17.8 \pm 1.7$ | $36.3 \pm 2.1$ | $30.1 \pm 3.7$ | $63.4 \pm 4.4$ | $57.3 \pm 1.0$ |
| EEGPT-Huge | ✓ | $44.7 \pm 4.2$ | **$20.7 \pm 2.3$** | $38.7 \pm 1.9$ | $32.3 \pm 2.7$ | $65.7 \pm 2.6$ | $59.1 \pm 1.3$ |
| EEGPT-Giant | ✓ | **$45.5 \pm 2.3$** | $19.9 \pm 1.9$ | **$41.3 \pm 1.5$** | **$33.9 \pm 1.4$** | **$67.2 \pm 3.3$** | **$60.4 \pm 1.8$** |

| Method | One Model? | Mental Workload | | Sleeping Stage | | Cross Modality | |
|---|---|---|---|---|---|---|---|
| | | EEGMat | STEW | EDF | HMC | IMG | SPE |
| *Specialist Models w/o pre-train* | | | | | | | |
| EEGNet | ✗ | $60.0 \pm 8.7$ | $52.3 \pm 17.6$ | $84.0 \pm 4.4$ | $54.5 \pm 8.7$ | $38.1 \pm 5.1$ | $52.2 \pm 1.4$ |
| TSception | ✗ | $50.3 \pm 1.2$ | $63.8 \pm 13.0$ | $68.6 \pm 4.5$ | $36.4 \pm 9.8$ | $31.3 \pm 3.0$ | $55.3 \pm 8.4$ |
| Conformer | ✗ | $49.8 \pm 1.1$ | $65.7 \pm 16.6$ | $67.4 \pm 3.5$ | $43.5 \pm 7.6$ | $35.0 \pm 3.9$ | $54.8 \pm 4.3$ |
| LGGNet | ✗ | $50.2 \pm 1.1$ | $46.7 \pm 12.5$ | $68.6 \pm 4.5$ | $17.0 \pm 9.5$ | $34.5 \pm 3.5$ | $52.4 \pm 5.8$ |
| *Specialist Models w/ pre-train* | | | | | | | |
| BIOT | ✗ | $50.2 \pm 1.1$ | – | – | – | – | $53.4 \pm 4.9$ |
| LaBraM | ✗ | $50.4 \pm 1.3$ | $52.5 \pm 12.4$ | $69.3 \pm 3.8$ | $39.4 \pm 9.4$ | $27.4 \pm 2.4$ | $50.9 \pm 1.4$ |
| *Generalist Models* | | | | | | | |
| EEGPT-Base | ✓ | $66.0 \pm 8.6$ | $63.2 \pm 10.6$ | $85.2 \pm 3.4$ | $65.5 \pm 4.0$ | $38.1 \pm 1.9$ | $58.2 \pm 2.6$ |
| EEGPT-Large | ✓ | $69.0 \pm 3.5$ | $65.4 \pm 10.1$ | $89.0 \pm 2.2$ | $66.8 \pm 3.5$ | $39.2 \pm 2.5$ | $60.4 \pm 2.9$ |
| EEGPT-Huge | ✓ | $70.7 \pm 6.2$ | $68.5 \pm 12.8$ | **$91.2 \pm 4.7$** | $68.2 \pm 1.3$ | $40.6 \pm 2.1$ | $60.3 \pm 3.5$ |
| EEGPT-Giant | ✓ | **$72.0 \pm 8.4$** | **$70.7 \pm 11.9$** | $90.6 \pm 1.9$ | **$70.3 \pm 2.2$** | **$41.5 \pm 1.7$** | **$61.6 \pm 2.4$** |

## 3.3 ABLATION STUDY

In this section, we conduct a detailed ablation analysis of the proposed training recipe. It is important to note that the findings are consistent across models of four different scales. Due to space limitations, we uniformly present the numerical results based on EEGPT-Large.

***Scaling law for model size preliminarily emerges.*** Figure 4 (a) compares the convergence curves of the autoregressive reconstruction loss across Base, Large, Huge, and Giant models. The results indicate that as the number of model parameters increases, the fit to the pre-training data improves,

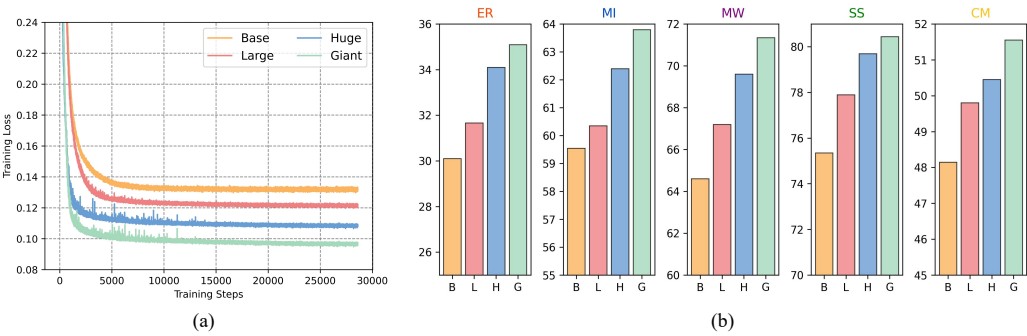

**Figure 4:** Scaling laws for model size: (a) Pre-training loss curves of EEGPT with varying parameter scales; (b) Performance of EEGPT on 5 downstream tasks across different parameter scales.

which is directly reflected in the final converged loss values. It largely indicates that models of a larger scale have absorbed more prior knowledge. We analyze the average performance of the four models across five tasks, as shown in Figure 4 (b). For all tasks, a positive correlation between performance and model size is evident, suggesting that larger models can effectively transfer more pre-training knowledge to a wide range of downstream tasks. This represents the first effective exploration and validation of the scaling law for autoregressive models in the EEG domain. We believe that with further increases in training scale, autoregressive architectures may exhibit enhanced generalization and versatility for EEG analysis.

***Scaling law for training data preliminarily emerges.*** In this section, we delve into another critical dimension: the scaling laws of training data. For our analysis, we randomly shuffle 1B tokens designated for pre-training and distribute them into five groups: 0B, 0.25B, 0.5B, 0.75B, and 1B tokens. Notably, the group with 0B tokens represents the absence of pre-training. We conduct pre-training across these varied data volumes. After freezing these pre-trained models, we perform multi-task fine-tuning, keeping training steps and settings consistent. The corresponding results are presented in Figure 5. As demonstrated in the figure, there are evident performance improvements across all five tasks as the volumes of pre-training data increase. These improvements are initially substantial but gradually taper off as data volumes expand. Similar patterns are also observed in the field of NLP (Kaplan et al., 2020). Given that the trend of performance improvement with increasing data has not yet diminished, we believe that by further expanding the dataset, EEGPT could achieve even better performance.

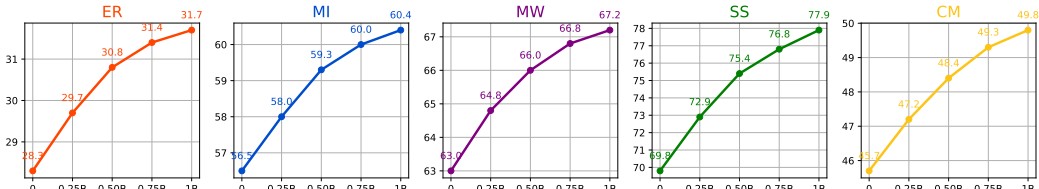

**Figure 5:** Scaling laws for data volume: As the size of training data increases, performance improvements are observed consistently across 5 tasks.

***Autoregression outperforms bidirectional masked pre-training.*** Current EEG pre-training models employ a masked signal reconstruction framework using bidirectional attention (Jiang et al., 2024; Yang et al., 2024b). In this framework, random segments of the signal are masked, and the resulting input is processed by an encoder that reconstructs these segments based on contextual information (*i.e.*, MAE). To enable a rigorous comparison between MAE and autoregressive (AR) modeling, we conduct an in-depth analysis using three distinct re-

**Table 4:** Comparison of the models with the pre-training objective of MAE *vs.* AR on 5 downstream tasks. Default setting is highlighted in blue .

| Method | Loss | ER | MI | MW | SS | CM |
|--------|------|------|------|------|------|------|
| MAE | cos | 26.4 | 56.6 | 62.9 | 70.4 | 43.2 |
| | $\ell_1$ | 27.8 | 60.8 | 61.6 | 73.2 | 45.3 |
| | $\ell_2$ | 29.7 | 59.7 | 63.3 | 74.8 | 47.0 |
| AR | cos | 28.6 | 59.1 | 63.8 | 72.2 | 45.9 |
| | $\ell_1$ | 30.0 | **61.2** | 65.0 | 74.5 | 48.6 |
| | $\ell_2$ | **31.7** | 60.4 | **67.2** | **77.9** | **49.8** |

construction loss functions: $\ell_1$, $\ell_2$, and cosine. For fairness, both pre-training paradigms utilize the same model architecture and parameter settings. We report related results in Table 4. For clarity, the standard deviations of the results presented have been omitted. The conclusions are twofold. First, for the three types of reconstruction loss, $\ell_2$ outperforms $\ell_1$, while cos shows the least efficacy. Second, irrespective of the distance metric employed, the AR architecture consistently outshines the MAE architecture, demonstrating a more than 2% average accuracy advantage. These findings align with and support existing research in the NLP domain (Radford et al., 2019; Achiam et al., 2023; Brown, 2020). Specifically, the unidirectional modeling task poses significant challenges, enabling the model to learn more robust representations. Additionally, AR effectively adapts to the temporal characteristics of EEG signals, capturing their patterns more directly and naturally.

***Mutual enhancement is observed among various tasks.*** We compare two downstream task training settings—joint multi-task training (default) and separate training—as shown in Table 5. For clarity, the standard deviations of the results presented have been omitted. In the separate training setting, each model is trained independently

**Table 5:** Comparison of the models with the settings of joint training *vs.* separate training on 5 downstream tasks. Default setting is highlighted in blue.

| Settings | ER | MI | MW | SS | CM |
|---|---|---|---|---|---|
| separate | 30.9 | 58.6 | 63.3 | 77.0 | 47.2 |
| joint | 31.7↑0.8 | 60.4↑1.8 | 67.2↑3.9 | 77.9↑0.9 | 48.8↑1.6 |

for each task, utilizing the same number of iterations and architectures as in the joint training scenario. Our observations indicate that models utilizing joint training consistently outperform those with separate training across all five tasks. Actually, for the two different datasets, the corresponding subgraphs share overlapping nodes. These shared nodes (*i.e.*, electrodes) provide a form of data augmentation that benefits both datasets. This augmentation is particularly important for tasks with limited samples. For instance, task MW has a total size of only 4K, whereas task SS reaches 42K. The accuracy benefits of joint training are more pronounced for task MW compared to task SS (*i.e.*, 3.9% for MW *vs.* 0.9% for SS). This enhancement suggests that despite originating from different tasks, signals from the same electrode exhibit shared patterns that can be effectively transferred. The introduction of the shared graph network effectively integrates and utilizes these shared patterns while also decoupling the differences between tasks. This phenomenon may provide an intriguing basis for future research on cross-task learning in EEG studies.

***Generalized representational ability even on unseen data.*** In this section, we explore a interesting conclusion regarding the transferability of ETE after autoregressive pre-training. Specifically, we utilize DREAMER (Katsigiannis & Ramzan, 2017), a dataset which is not included during the pre-training stage. This dataset comprises four categories formed by the $2 \times 2$ combinations of high and low valence and arousal dimensions. It is fed into ETE to obtain representations. The entire process does not involve shared graph networks or require additional training. Consistent with the pre-training stage, for each electrode, we extract the last token as the global representation for that electrode. These last tokens are then averaged across electrode dimension, resulting in the final representation for each signal. We employ t-SNE (Van der Maaten & Hinton, 2008) to visualize the underlying structures and patterns within these representations, as illustrated in Figure 6. The findings indicate that autoregressive pre-training demonstrates strong transferability even on unseen data, effectively clustering signal from different patterns/categories together.

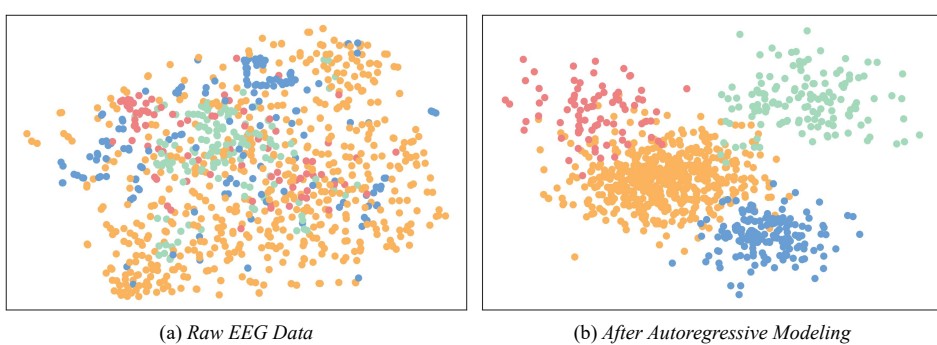

(a) *Raw EEG Data*        (b) *After Autoregressive Modeling*

**Figure 6:** t-SNE visualization comparison of the representation distributions before and after the autoregressive pre-trained Transformer. Different colors represent different categories.

## 4 CONCLUSION

In conclusion, we have presented EEGPT, the first generalist EEG foundation model designed to overcome the limitations of existing specialized EEG models. By introducing an electrode-wise modeling strategy, developing an autoregressive pre-training approach, and implementing a multi-task transfer learning paradigm with a learnable electrode graph network, EEGPT unifies diverse EEG datasets and captures the sequential and temporal dependencies inherent in EEG signals. Our model demonstrates superior performance across 12 benchmarks, showcasing its versatility and scalability. We hope that EEGPT will inspire further research and development in generalist EEG models.

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

## A   RELATED WORK

Despite the significant success of self-supervised pre-training in the fields of CV and NLP (Bai et al., 2024; Touvron et al., 2023), the potential of self-supervised pre-training in EEG remains underexplored. Specifically, existing work (Yang et al., 2024b; Jiang et al., 2024; Yi et al., 2024; Li et al., 2024) exhibits significant similarities in both the pre-training objective and the downstream task transfer paradigm. For pre-training objective, they predominantly employ the mask signal modeling (MAE) architecture. For instance, BIOT (Yang et al., 2024b) and MMM (Yi et al., 2024) adopt the channel and temporal embeddings to construct the EEG tokens for masked segments prediction. LaBraM (Jiang et al., 2024) utilizes a neural tokenizer to segment and encode EEG signals into discrete codes, and similarly predicts masked tokens from visible patches. However, the MAE architecture does not align with the temporal characteristics inherent in EEG signals. The models in the filed of NLP have shifted towards autoregressive models to better address similar temporal properties (Yang et al., 2024a; Touvron et al., 2023). Consequently, it is imperative to update the paradigm of self-supervised pre-training for EEG. Besides, these models encode EEG signals only for selected subsets of electrodes, which are lack of scalability and versatility. For downstream task transfer paradigm, they are still stuck to fine-tune separate models for each downstream task (Yang et al., 2024b; Jiang et al., 2024; Li et al., 2024) or even each subject (Yi et al., 2024), lacking the versatility required for an all-in-one model to multiple EEG tasks. Moreover, current pre-training models primarily focus on seizure epilepsy classification or emotion recognition tasks. In constrast, EEGPT aims for broader task coverage, enhancing both generalizability and adaptability.

## B   PRE-TRAINING DATASET DESCRIPTION

The detailed introduction of the datasets that we use for pre-training in our work and the data preprocessing procedure are as follows:

- **FACED** (Chen et al., 2023): FACED is a large finer-grained affective computing EEG dataset based on the discrete model, consisting of 30-channel EEG data recorded at a sampling rate of 250 or 1,000 Hz from 123 participants.

- **SEED** (Zheng & Lu, 2015): SEED is an emotion recognition dataset based on the discrete model, consisting of 62-channel EEG data recorded at a sampling rate of 1,000 Hz from 15 participants.

- **SEED-FRA** (Liu et al., 2022): SEED-FRA is an emotion recognition dataset based on the discrete model, consisting of 62-channel EEG data recorded at a sampling rate of 1,000 Hz from 8 French participants.

- **SEED-GER** (Liu et al., 2022): SEED-GER is an emotion recognition dataset based on the discrete model, consisting of 62-channel EEG data recorded at a sampling rate of 1,000 Hz from 8 German participants.

- **SEED-IV** (Zheng et al., 2018): SEED-IV is an emotion recognition dataset based on the discrete model, consisting of 62-channel EEG data recorded at a sampling rate of 200 Hz from 15 participants.

- **SEED-V** (Liu et al., 2021): SEED-V is an emotion recognition dataset based on the discrete model, consisting of 62-channel EEG data recorded at a sampling rate of 200 Hz from 16 participants.

- **THINGS-EEG-10Hz** (Grootswagers et al., 2022): THINGS-EEG-10Hz is a visual event-related potential (ERP) dataset that consists of 63-channel EEG data recorded at a sampling rate of 1,000 Hz from 50 participants. It includes 1,854 object concepts of 22,448 images from the THINGS (Hebart et al., 2019) stimulus set.

- **THINGS-EEG-5Hz** (Gifford et al., 2022): THINGS-EEG-5Hz is a visual event-related potential (ERP) dataset that consists of 122-channel EEG data recorded at a sampling rate of 1,000 Hz from 10 participants. It includes 1,854 object concepts of 16,740 images from the THINGS stimulus set.

- **IMG** (Private): IMG is a visual event-related potential (ERP) dataset that consists of 122-channel EEG data recorded at a sampling rate of 1,000 Hz from 32 participants. It includes five semantic categories of 2,500 images of the visual perception task.

For data preprocessing, all the EEG signals are resampled to 256 Hz. The signals are then filtered between 0.1 and 100 Hz and segmented into samples of four seconds. Each sample is further segmented into 25 tokens with an overlap rate of 0.875, while each token has 256 sampling points. Besides, we apply the z-score normalization. No further preprocessing or artifact correction methods are applied.

## C    MULTI-TASK DATASET DESCRIPTION

The detailed introduction of the datasets that we use for downstream tasks in our work are as follows:

- **DEAP** (Koelstra et al., 2011): DEAP is an emotion recognition dataset based on the dimensional model, consisting of 32-channel EEG data recorded at a sampling rate of 128 Hz from 32 participants. It describes emotion from two dimensions: valence and arousal, each comprising two categories—high and low. We employ a four-class classification based on these dimensions for the emotion recognition task.

- **FACED** (Chen et al., 2023): FACED is a large finer-grained affective computing EEG dataset based on the discrete model, consisting of 30-channel EEG data recorded at a sampling rate of 250 or 1,000 Hz from 123 participants. It contains data for nine emotion categories: amusement, inspiration, joy, tenderness; anger, fear, disgust, sadness, and neutral emotion. We employ a nine-class classification for the emotion recognition task.

- **SEED-IV** (Zheng et al., 2018): SEED-IV is an emotion recognition dataset based on the discrete model, consisting of 62-channel EEG data recorded at a sampling rate of 200 Hz from 15 participants. It contains data for four emotions: happy, sad, neutral, and fear. We employ a four-class classification for the emotion recognition task.

- **SEED-V** (Liu et al., 2021): SEED-V is an emotion recognition dataset based on the discrete model, consisting of 62-channel EEG data recorded at a sampling rate of 200 Hz from 16 participants. It contains data for five emotions: happy, sad, disgust, neutral, and fear. We employ a five-class classification for the emotion recognition task.

- **MIBCI** (Cho et al., 2017): MIBCI is a motor imagery dataset, consisting of 64-channel EEG data recorded at a sampling rate of 512 Hz from 52 participants. We employ a binary classification based on the left and right hands motor imagery.

- **BCI Competition IV-1 (BCIC4-1)** (Blankertz et al., 2007): BCIC4-1 is a motor imagery dataset which contains 59 channels of EEG data at a 100Hz sampling rate of 7 participants. We employ a binary classification based on the left or right hands and the both feet motor imagery.

- **EEGMat** (Zyma et al., 2019): EEGMat is a mental workload dataset comprising 23-channel EEG data recorded at a sampling rate of 500 Hz from 36 participants. The dataset includes two categories of states: rest and doing tasks. We employ a binary classification based on these states for the mental workload detection task.

- **STEW** (Lim et al., 2018): STEW is a mental workload dataset that includes 14-channel EEG data recorded at a sampling rate of 128 Hz from 45 participants. The dataset encompasses three levels of mental workload: low, medium, and high, allowing us to employ a three-class classification for the mental workload detection task.

- **EDF** (Kemp et al., 2000): The EDF dataset comprises 2-channel EEG data recorded at a sampling rate of 100 Hz from 78 participants. It includes five sleep stages: wake, N1, N2, N3, and movement, enabling us to conduct a five-class classification for the sleep stage.

- **HMC** (Alvarez-Estevez & Rijsman, 2021): The HMC dataset is a sleep dataset that comprises 4-channel EEG data recorded at a sampling rate of 256 Hz from 151 participants. It includes five sleep stages—wake, N1, N2, N3, and REM—facilitating a five-class classification for the sleep stage classification task.

- **IMG** (Private): IMG is a visual event-related potential (ERP) dataset that consists of 122-channel EEG data recorded at a sampling rate of 1,000 Hz from 32 participants. It includes five semantic categories of 2,500 images for a five-class classification of the visual perception task.

- **SPE** (Nguyen et al., 2017): SPE is a speech imagery dataset that consists of 64-channel EEG data recorded at a sampling rate of 256 Hz from 7 participants. It includes two types of words—long ("cooperate") and short ("in") for a binary classification of the cross-modality speech imagery task.

- **DREAMER** Katsigiannis & Ramzan (2017): DREAMER is an emotion recognition dataset based on the dimensional model that consists of 14-channels EEG data recorded at a sampling rate of 128Hz from 23 participants. It describes emotion from two dimensions: valence and arousal, each comprising two categories—high and low. We employ the four-class data for analysis.

## D    HYPERPARAMETER SETTINGS

In this section, we detail the training protocols of EEGPT. The specific hyper-parameter configurations for the Stage I: Autoregressive pre-training and the Stage II: Multi-task fine-tuning are reported in Table 6. The training time is based on 8 NVIDIA A800-80G GPUs

**Table 6:** Training hyperparameters for EEGPT of two training stage.

| Stage | Hyperparameter | Base | Large | Huge | Giant |
|---|---|---|---|---|---|
| Stage I | Lr | | | 1e-4 | |
| | Time | 3.2h | 7.6h | 11.2h | 19.8h |
| | Epoch | | | 3.0 | |
| | Precision | | | BF16 | |
| | Deepspeed | Zero2 | Zero2 | Zero2 | Zero3 |
| | LR Schedule | | cosine decay | | |
| | Warmup Ratio | | | 0.03 | |
| | Batch Size per GPU | | | 512 | |
| | Gradient Checkpoint | | | True | |
| Stage II | Lr | | | 1e-4 | |
| | Time | 0.3h | 0.7h | 1.6h | 2.9h |
| | Epoch | | | 10 | |
| | Precision | | | BF16 | |
| | Deepspeed | Zero2 | Zero2 | Zero2 | Zero2 |
| | LR Schedule | | cosine decay | | |
| | Warmup Ratio | | | 0.1 | |
| | Batch Size per GPU | | | 64 | |
| | Gradient Checkpoint | | | True | |

## E    BASELINE MODEL DESCRIPTION

The detailed descriptions of the six baseline models that we reproduce for comparison in this work are as follows:

- **EEGNet** (Lawhern et al., 2018): EEGNet is a compact convolutional neural network designed for EEG-based brain-computer interfaces. It leverages depthwise and separable convolutions to facilitate efficient feature extraction and classification.

- **TSception** (Ding et al., 2022): TSception is a multi-scale convolutional neural network designed for EEG emotion recognition, capable of learning discriminative representations across both time and channel dimensions. The model incorporates a dynamic temporal layer to effectively capture dynamic temporal and frequency representations, while an asymmetric spatial layer is employed to learn discriminative global and hemisphere representations.

- **Conformer** (Song et al., 2023): Conformer is a compact convolutional Transformer model designed for EEG classification, capable of encapsulating both local and global features.

It incorporates a convolutional module to effectively learn low-level local features while employing a self-attention mechanism to extract global correlations within the local temporal features.

- **LGGNet** (Ding et al., 2023): LGGNet is a neurologically inspired graph neural network designed for EEG representation learning. It effectively models the intricate relationships both within and between the brain's functional regions.

- **BIOT** (Yang et al., 2024b): BIOT is a self-supervised biosignal learning model that tokenizes biosignals of various formats into "sentences". It segments each channel separately into tokens and flatten the tokens to form "sentences". Through its MAE architecture, it can be pre-trained with unlabelled data.

- **LaBraM** (Jiang et al., 2024): LaBraM is a self-supervised EEG model that enables cross-dataset learning by segmenting the EEG signals into channel patches. It adopts the Vector-quantized neural spectrum prediction to train a neural tokenizer that encodes EEG patches into compact neural codes. Through its MAE architecture, it can be pre-trained with unlabelled data.

## F  NAME OF THE SUPPORTING ELECTRODES

The name of the supporting electrodes of our EEGPT are as listed in Table 7.

| PO12 | CCP2H | FFC5H | OI1 | PO7 | CPPZ |
|------|-------|-------|-----|-----|------|
| TP7 | PO2 | FC3 | FTT7H | PPO8 | CCP4H |
| P11 | FCC5H | FFC4H | FP1 | CPP2H | FFT7H |
| P1 | I2 | AFF6H | FZ | PO4 | FCC2H |
| F8 | FT9 | CP2 | AF3 | FCZ | POO11H |
| FPZ | F3 | P8 | FC2 | F1 | CCP3H |
| CP6 | PO1 | C1 | AFZ | C3 | CB1 |
| FTT8H | POO12H | TP9 | I1 | FP2 | POO10H |
| CPP1H | CPP4H | TTP8H | AFF5H | PO10 | POO9H |
| POO3 | CP5 | PO3 | FC6 | FTT9H | PPOZ |
| TPP5H | POO4 | CB2 | FT7 | CPZ | CP1 |
| PPO1 | CP3 | CCP5H | O2 | FCC1H | CP4 |
| FT8 | T9 | PO5 | P2 | P5 | POZ |
| FC1 | CPP3H | C5 | P9 | P10 | PO6 |
| FFT8H | CCP1H | C2 | POOZ | T7 | POO7 |
| FFC3H | F6 | FCCZ | TPP8H | F7 | P4 |
| P3 | AF8 | PPO2 | AF4 | FFC2H | FFC1H |
| P6 | F2 | C6 | P12 | TP10 | CZ |
| IZ | CCP6H | TP8 | PO11 | OI2 | FC5 |
| TTP7H | CPP5H | F5 | POO8 | CPP6H | OZ |
| PO9 | AF7 | PZ | O1 | FC4 | PO8 |
| F4 | FCC3H | T10 | P7 | FT10 | FCC4H |
| FCC6H | T8 | PPO7 | C4 | FFC6H | FTT10H |

**Table 7:** Supporting EEG electrodes.

