# OpenReview forum: "EEGPT: Unleashing the Potential of EEG Generalist Foundation Model by Autoregressive Pre-training"
_ICLR.cc/2025/Conference — ICLR 2025 Conference Withdrawn Submission_

### Official Review · Reviewer_wTwy · 2024-10-21

**Soundness:** 3
**Presentation:** 2
**Contribution:** 3
**Rating:** 5
**Confidence:** 4

**Summary:**

This paper proposes a construction method for a large EEG model, organizing model inputs by channel. It employs a "next sentence prediction" approach for pretraining the large model and introduces a multi-task fine-tuning method to adapt to downstream tasks simultaneously. During fine-tuning, a graph network is used to adapt to different datasets and capture the relationships between channels. The approach achieves state-of-the-art performance on five tasks across twelve datasets.

**Strengths:**

1.Innovation: Applies the unsupervised "next sentence prediction" pretraining method from natural language processing to the construction of a foundational EEG model, including attempts to apply and compare its effects with mask-based reconstruction methods; also validates the advantages of multi-task fine-tuning.
2.Extensive experimentation: Validates the model’s effectiveness across multiple datasets across five tasks, including comparisons with mask-based reconstruction methods.

**Weaknesses:**

1.Innovation: The proposed pretraining method does not extend beyond the two tasks used in Bert [1] — mask-based reconstruction and next sentence prediction. While Bert employs both tasks simultaneously in training language models, and LaBraM[2] utilizes the mask-based reconstruction approach, this paper explores the other method without attempting to combine both approaches, limiting its innovation in large model pretraining methods.
2.Method design: The construction of large models generally focuses more on unsupervised pretraining methods rather than the benefits of multi-task fine-tuning across different datasets. This requires the assumption of sufficient annotated data, and the high cost of annotation may make this approach impractical for real-world tasks.
3.Clarity of description: When discussing the unification of different dataset formats for input construction, the paper mentions organizing inputs by channel and adding a special channel identifier for each sample. However, significant dimensional discrepancies among different datasets and how to mitigate these discrepancies through unified input construction (possibly using LaBraM’s method to construct dense inputs) are not clearly explained. Further clarification is recommended.

[1]Jacob Devlin, Ming-Wei Chang, Kenton Lee, Kristina Toutanova. BERT: Pre-training of Deep Bidirectional Transformers for Language Understanding. arXiv. 2019.
[2]Wei-Bang Jiang, Li-Ming Zhao, and Bao-Liang Lu. Large brain model for learning generic representations with tremendous eeg data in bci. arXiv preprint arXiv:2405.18765, 2024.

**Questions:**

1.Why are only nine datasets used for pretraining while thirteen are used for fine-tuning, with some datasets overlapping between the two sets? What is the rationale behind this selection?
2.Why is the performance so low across all datasets on the emotion recognition task, e.g., only a 20% accuracy on the FACED dataset, suggesting that the method may be impractical for real-world applications?
3.It is recommended to include ablation experiments for fine-tuning from scratch, i.e., training datasets directly using multi-task learning methods without using the pretrained large model, to highlight the benefits of the pretraining model.

---

### Official Review · Reviewer_muJ4 · 2024-10-29

**Soundness:** 2
**Presentation:** 2
**Contribution:** 2
**Rating:** 3
**Confidence:** 5

**Summary:**

The submitted work addresses the topic of model generalization for electroencephalography (EEG) data.
This is a highly relevant problem for EEG-research with potentially high impact on various EEG-based neurotechnology applications.
The fundamental problem is that current models generalize poorly across individuals (e.g., subjects) and tasks.
To address this problem, the submitted work proposes to pool diverse EEG datasets comprising different tasks and electrode setups, and learn an autoregressive foundation model.
Specifically, the authors suggest a two-phase approach. In the first phase, they propose to pre-train an electrode temporal encoder (ETE) network that converts single-electrode time-series data (and a learnable electrode embedding) with a standard auto-regressive loss.
In the second phase, the weights of the ETE module are frozen.
To integrate spatial information, the authors propose a task-shared electrode graph (TEG) network that defines static (per-task) fully-connected graphs whose features are a combination of learnable embeddings and transformed summary tokens obtained from the ETE network.
The network integrates spatial information and outputs a final embedding that is passed to task-specific decoding heads.

The authors evaluate their approach with 4 network variants ranging from 1.5 M to 1.1 B parameters.
They compare their approach to several baseline models.
The considered baselines included neural architectures that were proposed to train specialist models for single-tasks or self-supervised pre-training.

All models were evaluated across 12 datasets, each associated with a distinct task category (including emotion recognition, sleep stating, ...). The authors report that their proposed approach (pre-training + multi-task fine-tuning) achieves the best results with a margin of 5+ percent points across categories.
In ablation studies, they report that the models with more parameters performed better on the pre-training and downstream task, that more data resulted in better generalization, and the auto-regressive approach had an edge over a masked-auto-encoder approach.
Additionally, they report that the tasks with little data (e.g., mental workload) benefit most from the multi-task fine-tuning phase. Lastly, they provide qualitative results in the form of t-SNE embeddings that show an emerging cluster structure according to the task labels for a previously unseen emotion recognition dataset.

**Strengths:**

### Originality

I appreciate the authors effort to collect and combine diverse EEG datasets.
Altogether, the strength of this contribution is the combination of existing ideas for an established application problem (e.g., generalization for EEG neurotech)

### Quality

Key to the success of EEG foundation models is to combine temporal/spectral/spatial integration in the presence of diverse hardware and electrode configurations.
The presented empirical results indicate that this contribution proposes a suitable solution for the problem.
Additionally, the ablation study touches on several relevant directions.

### Clarity

Thanks to the introduction and Figures 2 and 3, the problem definition and conceptual overview of the approach are clear.

### Significance

The study confirms that neural network scaling laws also apply to EEG data, suggesting that pooling diverse datasets and learning models with millions to billions of parameters can yield performance gains. Whether this strategy will be enough to elevate the performance of EEG-based neurotech remains questionable.

**Weaknesses:**

### Tokenization strategy
My biggest concern about the submission is the lack of investigations on EEG tokenization.
The main text does not provide any information about EEG pre-processing and harmonization, leaving the reader puzzled about the dimensions $T$ and $C$ as defined in section 2.1 , and whether the data were harmonized (e.g., to a common sampling rate).
After some digging, I found a short paragraph on data preprocessing in the Appendix on page 16 (but no reference to it in the main text!).
This paragraph contains important information to understand the approach of the method and should, therefore, be move to the main text.

Without any justification or reference to prior work, the authors decided to use substantially overlapping short temporal windows (i.e., 1 second) to segment slightly longer windows of EEG time-series data (i.e., 4 seconds) into 25 tokens (i.e., T = 25), each containing 1 second of data at a rate of 256 Hz (i.e., C = 256).
Since auto-regressive pre-training is one of the main contributions of this paper, the authors should provide additional motivation for the particular choice of hyper-parameters and ideally run ablation studies that investigate different hyper-parameter choices (e.g., length of the long and short windows as well as the overlap).

### Clarity of multi-electrode and multi-task fine-tuning approach
The authors describe their multi-electrode integration approach as a graph that is task-specific.
Unfortunately, the provided description is insufficient.
A graph typically comprises vertices (or nodes) and edges.
The author's merely define the node features but do not clearly introduce how they determine edges and their values (based on equation (9), I assume that they use binary edge-weights based on the availability of an electrode in a sample $z_j$).
Additionally, in equation (8) the authors state that $ \mathrm{diag}(z_j) $ converts $z_j$ into a matrix. However, $z_j$, defined in equation (7), is already a matrix.
Overall, the clarity of the this section would benefit a lot from proper function definitions that include the function's image and domain.

### Organization
The authors spend a considerable amount of text on repeatedly highlighting their perspective of the submission's contributions. I think that the authors should rather expand on the methods description and experimental results.

**Questions:**

### Introduction
There are some relevant works that I think should be included:
- The authors in [1] compared autoregressive (GPT2) with masked autoencoder (BERT) networks with regard to generalizable latent representations that retain task-relevant information in functional magnetic resonance imaging (fMRI) data. The work also reports superior results for the autoregressive approach.
- Two recent papers [2,3] also proposed to use an electrode-wise modeling strategy. The authors in [2] combined this module with a dynamic graph neural network in a supervised learning scenario, while [3] used it in a JEPA-style SSL scenario.


### Methods
- Line 165: please define the index $i$, and change $x$ to $x_i$
- I strongly recommend to move the paragraph about data pre-processing (in appendix B) before section 2.1 and expand it.
- Please define sources in line 177. I think you should potentially introduce additional indices or mappings in your notation that can be used to clearly associate individual samples to a specific task/source dataset. The current (lack of) notation is confusing.
- Notation of $\mathcal{R}(\cdot)$ in (1) is inconsistent with line 189 (and lines 268, 270)
- Either use $x_i^e$ or $\mathrm{x}_i^e$
- Lines 198-211: I think you should cite the original GPT-family and transformer papers in this paragraph.
- Which normalization layer did you use?
- Did you use position encoding to encode the temporal order? If so, which type of position encoding did you apply?
- Coding the application of the mask $M$ with the $+$ symbol is highly irregular.
- Inconsistency between Figure 3 and line 252. I do not see any graphical representation of the learnable special token $c$ in Figure 3.
- To be consistent with equation 3, I think you should also concatenate the electrode embedding in equation 6.
- Please properly define the graph $\mathcal{G}$ in line 270 (see weaknesses comment for details).
- Equation 8 is difficult to follow. Please define the dimensions of the items. In my understanding you want to express that you add the learnable node features of $\mathcal{G}$ with the embeddings $z_j$ for all electrodes that are present in sample $z_j$, right?
- Please add a more detailed description of the MLP decoding heads as well as the employed loss function to train the model in the fine-tuning phase.
- Lines 293 to 298: important information is missing. Which normalization layers did you use? What was the loss function? To improve clarity, I think you should include a detailed listing/visualization of the layers/blocks that you used within your module in the appendix.

### Experiments
- Line 325 (and following): please provide a direct reference to the specific appendix.
- How did you segment the data for the pre-training phase (i.e., did you just extract non-overlapping 4-second periods from the continuous data? Or did you use event markers and extract 4-second epochs around those)? Was the procedure similar for the fine-tuning phase? How about model evaluation with labeled data?
- How did you allocate samples to mini-batches? Randomly or did you combine smaller batches of different datasets?
- Why is there no result for BIOT and BCIC4-1 in Table 4?
- Figure 4: include labels for the axes.
- Line 476: in the methods you describe that you use the MSE loss, but here you use the term $l_2$ loss. They are not exactly the same.
- Tables 4/5: please also report the standard deviations. A measure of variability is highly important to properly assess effect strengths.
- Results in Figure 6 are difficult to interpret.
  Please clarify how you generated the plot in Figure 6a. Did you plot the raw EEG data after/before pre-processing or the output of ETE with random initializations? I think you should also include representations obtained with pre-trained BIOT and/or LaBraM.


### Wording, Grammar and Organization
- line 56: the comparison of EEG as the language of the brain seems far fetched. We know that EEG captures activity of large-scale brain networks - primarily in the form of dipoles forming along large-networks of cortical pyramidal neurons.
- Figure 1 and 2 are not referenced in the main text.
- line 127: what do you mean with "seqeuntial and temporal dependencies"?
- line 130: "To the best of our knowledge, It ..." -> "..., it, ..."
- line 134: there are grammar issues
- line 145: the sentence is difficult to follow
- line 334: grammar issues
- line 506: "..., we explore a interesting ..." -> "..., we explore an interesting ..."

### References

[1] A. Thomas, C. Ré, and R. Poldrack, “Self-Supervised Learning of Brain Dynamics from Broad Neuroimaging Data,” in Advances in Neural Information Processing Systems, S. Koyejo, S. Mohamed, A. Agarwal, D. Belgrave, K. Cho, and A. Oh, Eds., Curran Associates, Inc., 2022, pp. 21255–21269.

[2] S. Tang et al., “Modeling Multivariate Biosignals With Graph Neural Networks and Structured State Space Models,” in Conference on Health, Inference, and Learning, PMLR, 2023, pp. 50–71.

[3] Guetschel, Pierre, Moreau, Thomas, and Tangermann, Michael, “S-JEPA: TOWARDS SEAMLESS CROSS-DATASET TRANSFER THROUGH DYNAMIC SPATIAL ATTENTION,” in Proceedings of the 9th Graz Brain-Computer Interface Conference, Graz, Austria: Verlag der Technischen Universität Graz, 2024. doi: 10.3217/978-3-99161-014-4-003.

---

### Official Review · Reviewer_ttPg · 2024-11-03

**Soundness:** 1
**Presentation:** 3
**Contribution:** 2
**Rating:** 3
**Confidence:** 5

**Summary:**

The authors introduce a model for EEG decoding on multiple tasks simultaneously. The architecture starts with 1) a shared transformer-based module (the Electrode Temporal Encoder) which encodes each channel independently, then is followed by 2) a shared graph network which combines the information across channels and finally, 3) task-specific MLPs which return predictions.
Twelve EEG datasets are used from different fields: emotion recognition, motor imagery, mental workload, sleep stage and cross modalities.
Training is done in two steps:
1. This electrode temporal encoder is pre-trained alone in an unsupervised manner on a next-token prediction task
2. The whole network is finetuned on the (supervised) classification tasks.
Validation and testing are done on subjects that were left out from the different datasets for training.

**Strengths:**

The article is overall well-written and easy to follow. The figures and tables are sufficiently clear (except fig. 1, see below) and use a unified colour code for the datasets and model sizes which improves readability.
The authors conducted a scalability study of their model, which is still relatively rare in BCI.
I recognise the important effort that was made in reproducing previous studies. This is not an easy task and it highlights the necessity of using frameworks such as the MOABB library (http://moabb.neurotechx.com) which would have avoided such a redundant task.
Finally, the authors promised to publish their checkpoints, which is always good for reproducibility and not done enough in the BCI field.

**Weaknesses:**

Major
- The evaluation method is problematic and does not support the main claim of the article, which is that the authors introduce a “generalist EEG foundation model”. As stated in lines 660-664, the authors “adopt a cross-subject paradigm”, and the validation and test sets contain different subjects but from the same datasets as for training. Therefore, the model does not realise a cross-task transfer but rather a simple cross-subject transfer learning scenario (augmented with examples from other tasks). The variability between different subjects is not comparable to that of different datasets or even different decoding tasks. In this article, the model was trained on subjects doing the exact same tasks from the exact same datasets as during testing. Therefore, it is impossible to predict from these results how the model will behave on new tasks from new datasets.
- The authors make multiple strong claims (all starting with “the first…”) which are either overstatements, if not wrong. The presentation of this work raises concerns about its scholarly rigour and gives the impression of being more like a product promotion than a scientific publication. The  claims are the following:
    - “first electrode-wise modeling strategy”. See for example:
        - Yang et al. (2023) http://arxiv.org/abs/2305.10351
        - Guetschel et al. (2024) https://doi.org/10.3217/978-3-99161-014-4-003
        - Li et al. (2024) https://doi.org/10.1109/TNSRE.2024.3357863
    - “first autoregressive model”. See Banville et al. (2021) https://www.doi.org/10.1088/1741-2552/abca18. Moreover, multiple articles already explored masking-based SSL strategies, which are not strictly speaking “autoregressive” but still very similar. See for example:
        - Foumani et al. (2024) https://doi.org/10.1145/3637528.3671600
        - Yang et al. (2023) http://arxiv.org/abs/2305.10351
    - “first generalist EEG model for multi-task compatibility and synergism”. See Yang et al. (2023) http://arxiv.org/abs/2305.10351. moreover, multiple works already explored cross-task transfer learning in BCI:
        - Aristimunha et al. (2023) https://arxiv.org/abs/2308.02408
        - Guetschel et al. (2023) https://arxiv.org/abs/2311.16109
- Some results appear significantly below the state-of-the-art. Some variability might be explained by differences in the evaluation settings but the differences reported (10%) are non-negligible. Unfortunately, I am not familiar with all datasets but here are those I know well:
    - MIBCI. Zhao et al. (2020) obtained 76.5% accuracy in a similar cross-subject configuration, which is 9.3% above the reported score and 13.2% above the best specialist model https://doi.org/10.1155/2020/7285057
    - Seed-IV. Li et al. (2018) reported that a simple SVM (one of their baselines) obtained 52.8% accuracy in a similar cross-subject transfer setting, which is 11.5% below the reported score and 18.1% below the best specialist model https://doi.org/10.1007/978-3-030-04221-9_36

Minor
- The name EEGPT is probably not the best choice as two articles already used that name for their architectures:
    - https://neurips.cc/virtual/2024/poster/93793
    - https://arxiv.org/abs/2401.18006
- Figure 1 is very difficult to read. The axes scale is defined by the blue curve therefore, the tickmarks land on random values. I understand that the scores on the different datasets are not comparable together but you should at least scale the axes such that the tickmarks land on integer values, not floats with two decimals. Also, it would help the reader to add the range of each axis next to its name.
- line 86: the claim that MAE “have inevitable limitations in capturing the sequential and temporal dependencies” is not motivated. Research actually showed that they can. See Chien et al. (2022) http://arxiv.org/abs/2211.02625.
- The choice of motor imagery datasets seems rather arbitrary. Cho 2017 and BCI Competition IV 1 are not commonly used for benchmarking. Instead, I would have recommended:
    - Very common benchmark: dataset B from 2008 BCI competition http://moabb.neurotechx.com/docs/generated/moabb.datasets.BNCI2014_004.html
    - More recent and one of the largest motor imagery datasets: Stieger et al. (2021) http://moabb.neurotechx.com/docs/generated/moabb.datasets.Stieger2021.html

**Questions:**

Please, refer the the weaknesses section.

---

### Official Review · Reviewer_ZzEm · 2024-11-04

**Soundness:** 3
**Presentation:** 4
**Contribution:** 2
**Rating:** 5
**Confidence:** 4

**Summary:**

This paper presents EEGPT, an innovative EEG foundation model leveraging univariate fine autoregressive next-token prediction and multivariate fine-tuning to address the challenges of diverse data formats, outdated pre-training approaches, and limited transfer learning techniques in EEG research.

**Strengths:**

The paper is well-written and presents a highly engaging approach by utilizing a wide array of EEG datasets, showcasing impressive results that push the boundaries of current EEG modeling.

**Weaknesses:**

Additional references should support the reported results, as certain datasets appear to have higher state-of-the-art (SOTA) values than those presented.
Sharing anonymized code would enhance reproducibility and allow for broader validation of the model.
Lastly, an in-depth discussion on how the model captures covariances between variables is crucial, given its importance in the EEG literature. This would provide clarity on EEGPT’s handling of multivariate dependencies.

**Questions:**

The paper would benefit from a clearer explanation of the tokenization strategy, as understanding this process is key to evaluating the model’s foundation.

---

### Note · Authors · 2024-11-13

I have read and agree with the venue's withdrawal policy on behalf of myself and my co-authors.